# Rethinking the Value of Training-Free Structured Pruning of LLMs

**Nahush Lele**[*,1,2], **Arnav Chavan**[*,3], **Aryamaan Thakur**[1,2], **Deepak Gupta**[4]
[1]*Indian Institute of Technology (ISM), Dhanbad, India*
[2]*Nyun AI, India*
[3]*Amazon Lab126*
[4]*Transmute AI Lab (Texmin Hub)*

**Reviewed on OpenReview:** *https://openreview.net/forum?id=7KkytYYhMv*

## Abstract

This paper investigates the effectiveness of training-free structured pruning techniques for Large Language Models (LLMs), with a particular focus on depth and width pruning strategies. Through an extensive empirical evaluation across a diverse range of tasks, datasets and modalities, we reveal critical limitations in current pruning methods. While some tasks exhibit minimal performance degradation, others face significant deterioration, even at low pruning rates, contradicting prior findings that often rely on selective benchmarks. Our analysis also finds that depth pruning, despite its simplicity, usually outperforms the more granular width pruning approaches in maintaining downstream task performance. Our findings highlight that existing evaluations of pruned LLMs often overstate their effectiveness due to incomplete or limited evaluation tasks, necessitating a critical reassessment of the true value of pruning and emphasizing the need to explore more robust pruning algorithms.

## 1 Introduction

Due to their large size and complexity, Large Language Models (LLMs) face several challenges in deployment for real-world applications (Thompson et al., 2022). This issue has sparked significant research focused on creating lightweight LLMs (Mehta et al., 2024; Thawakar et al., 2024; Abdin et al., 2024) or compressing existing models (Gu et al., 2024; Sreenivas et al., 2024) to enable efficient use in resource-constrained settings, such as mobile devices, edge servers, and embedded systems. The primary goal of these efforts is to balance model efficiency with minimal performance degradation, ensuring that LLMs remain functional across various applications while reducing their resource demands.

To achieve this balance, multiple compression techniques such as pruning (Ma et al., 2023; An et al., 2023a), quantization (Lin et al., 2024a; Frantar et al., 2023), matrix decomposition (Lin et al., 2024b; Chavan et al., 2024a;b) and knowledge distillation (Sanh et al., 2020; Jiao et al., 2020; Gu et al., 2024) have been explored that modify the structure and operation of LLMs to reduce their size and computation needs (Chavan et al., 2024c). Structured pruning (Molchanov et al., 2016), in particular, has emerged as a key method for compressing LLMs by selectively removing model components—such as layers or neurons—without extensive retraining. Unlike unstructured pruning (Han et al., 2015a), which targets individual weights, structured pruning offers the advantage of removing entire groups of parameters, thus yielding greater efficiency gains and simplifying deployment. However, traditional structured pruning methods often rely on retraining

---

[*]Equal contribution. Work done while Arnav was CTO at Nyun AI.

or fine-tuning to recover any performance loss, which can be computationally costly and time-consuming, especially for very large models (Frankle et al., 2020). In response to these limitations, there is a growing interest in training-free structured pruning techniques, which avoid the need for additional retraining (Ma et al., 2023; An et al., 2023a; Kim et al., 2024; Men et al., 2024). Despite their potential to enable faster deployment, training-free methods remain underexplored, particularly in the context of how they impact model performance and generalization across various downstream tasks.

In this study, we rigorously investigate training-free structured pruning techniques in the context of LLMs, examining the performance implications of both depth pruning (removing entire layers) and width pruning (removing neurons or channels) across diverse tasks. Our contributions are threefold:

- *Comprehensive Evaluation of Training-Free Structured Pruning of LLMs*: We present an extensive empirical analysis of recent training-free structured pruning methods applied to LLMs, extending the evaluation to additional tasks and datasets beyond those previously reported. This broader assessment allows us to identify patterns and limitations in pruning that were not apparent in prior studies.

- *Critical Benchmarking of Pruning Effects*: Our findings reveal that even at minimal pruning rates, significant performance degradation can occur on critical benchmarking tasks, which contrasts with prior work that often reports results on selectively curated benchmarks. By evaluating the impact of pruning across a representative set of standard benchmarks, we highlight the potential risks and limitations associated with training-free structured pruning, underscoring the need for caution when applying these methods broadly.

- *Insights into various Pruning Strategies*: We provide novel insights into the prunability of LLMs across both depth and width pruning regimes. Our analysis reveals distinct pruning dynamics across model sizes, challenging conventional assumptions by showing that depth pruning, despite its simplicity, can sometimes outperform width pruning in maintaining task-specific performance. We also explore the bias and fairness implications of pruning, highlighting ethical considerations that have yet to be fully addressed in existing research.

Our findings raise important questions about the broader applicability of current pruning techniques and contributes to the growing understanding of efficient LLM compression, providing actionable insights for researchers and practitioners focused on deploying these models across resource-limited environments.

## 2 Related Work

*Model Compression Techniques for LLMs* have played an important role in enabling the deployment of LLMs on resource constrained hardware. Traditional methods, such as quantization (Han et al., 2015b), reduce the precision of weights and activations thereby reducing the memory and computational requirements (Lin et al., 2024a; Frantar et al., 2023). Matrix decomposition techniques, such as low-rank factorizations, approximate large weight matrices with smaller, more efficient representations, and have been utilized in LLMs to reduce the number of parameters while maintaining model quality (Chavan et al., 2024a;b). Knowledge distillation, aims at transferring knowledge from a larger teacher model to a smaller student model, allowing the creation of smaller networks which inherently use fewer resources (Gu et al., 2024; Sreenivas et al., 2024). While these approaches have demonstrated effectiveness in reducing the size and complexity of LLMs, they often require additional training or fine-tuning steps, which can be computationally costly, especially for very large models.

Structured pruning has gained prominence as a particularly effective technique for reducing model size and improving practical efficiency (Molchanov et al., 2016). By removing entire structures—such as channels, or layers—structured pruning maintains a more interpretable and hardware-friendly model structure, unlike unstructured pruning, which involves removing individual weights and can lead to sparse matrices that are challenging to deploy (Li et al., 2017). Studies have shown that structured pruning can effectively compress models without significant performance loss, especially when combined with fine-tuning (He et al., 2017a).

*Training-Free Structured Pruning:* While structured pruning methods have shown success, they often rely on retraining to recover lost performance, a process that is resource-intensive for large models. To mitigate this, recent research has explored training-free structured pruning techniques that aim to simplify models without requiring additional training steps. Approaches in this domain include pruning based on weight magnitudes, neuron activations, or structural properties of the model (Ma et al., 2023; Dery et al., 2024; Li et al., 2024; An et al., 2023a).

*Bias and Fairness in Pruned Models:* The impact of model pruning on bias and fairness has become an area of growing concern, especially as LLMs are increasingly applied to sensitive domains. Studies have shown that pruning can unintentionally affect a model's learned representations, potentially amplifying biases or reducing performance on underrepresented groups (Hooker et al., 2020; Ramesh et al., 2023). Fairness-aware pruning strategies have been proposed to mitigate these effects, typically by incorporating fairness constraints during the pruning process (Dai et al., 2023; Lin et al., 2022). However, little work has examined these impacts specifically for pruning, leaving an important gap in the literature. In our study, we extend the analysis of training-free pruning techniques to include the effects on bias and fairness, providing novel insights into the ethical implications of model compression.

## 3 Background

This study investigates the impact of training-free structured pruning on large language models (LLMs). We evaluate two main pruning strategies—depth and width pruning—across a variety of LLMs and downstream tasks. Our analysis spans both standard and lesser-explored tasks, emphasizing the importance of comprehensive task coverage when assessing pruning effects. Additionally, we examine the influence of pruning on model fairness and bias, providing a holistic view of its implications on model performance, resource utilization, and equitable treatment across demographic groups.

### 3.1 Training-Free Structured Pruning

Structured pruning has emerged as an effective technique for compressing deep neural networks by selectively removing parts of the model in an interpretable manner. Unlike **unstructured pruning**, which removes individual weights and can lead to sparse connections, structured pruning targets groups of parameters, such as channels, or entire layers. This approach simplifies the final model structure, making it more compatible with hardware accelerators and easier to deploy in real-world scenarios (Liu et al., 2018; He et al., 2017b). However, structured pruning presents several challenges, particularly structured pruning is followed by resource intensive fine-tuning in order to recover lost performance. This is especially problematic in the case of billion-scale LLMs where retraining based on full-model gradients is limited to large enterprises (Liu et al., 2018). Training-free structured pruning bypasses the need for retraining by directly pruning the model without any subsequent optimization steps, making it highly suitable for scenarios with limited computational resources. They rely on intrinsic properties of the model, such as weight magnitudes and activation fluctuations, to identify and remove less critical components, often substituting the pruned components with additional bias terms (An et al., 2023a). Training-free structured pruning is often complemented with parameter-efficient fine-tuning (PEFT) techniques (Hu et al., 2021; Lian et al., 2022), however the improvements with PEFT are inferior as compared to full-finetuning Ramesh et al. (2023). Nevertheless, training free structured pruning reduces memory and computational costs, enabling deployment on resource-constrained devices or environments where retraining infrastructure may not be available.

However, training-free pruning comes with significant trade-offs. Because these methods do not allow the model to adapt post-pruning, they can lead to more pronounced performance degradation, especially for complex tasks that rely on nuanced representations. Without the corrective phase of fine-tuning, training-free pruned models may experience reduced accuracy, affecting their reliability across diverse applications. Unlike smaller models, LLMs have extensive feature hierarchies and complex dependencies that may be disrupted by pruning, particularly without the restorative benefits of fine-tuning. As such, while training-free pruning provides efficiency, it may compromise task-specific performance, highlighting a need to balance compression gains with acceptable accuracy loss.

### 3.2 Pruning Techniques

Our chosen pruning approaches encompass both depth and width pruning techniques independently, targeting different aspects of model structure. Depth pruning involves the selective removal of entire model blocks, guided by an activation similarity based importance criterion (Men et al., 2024; Gromov et al., 2024; Jha et al., 2024; Liu et al., 2023), whereas width pruning leverages activation fluctuations to evaluate and adaptively compress individual weight columns within model layers (An et al., 2023a; Ma et al., 2023). More details on selected pruning methods are provided in Appendix.

**Depth Pruning:** We adopt activation similarity based depth pruning method (Men et al., 2024; Gromov et al., 2024). ShortGPT calculates the cosine similarity between the input and output embeddings of the $i$th block, using cosine similarity as a global pruning criterion across model layers. The rationale behind this approach is that if a block's input and output features show minimal difference—as indicated by high cosine similarity—then that block contributes less significantly, relative to other blocks with lower cosine similarity.

**Width Pruning**: Width pruning involves selectively pruning heads and channels from the attention and MLP layers respectively within each module. In the attention layer, pruning is performed at the granularity of heads, meaning that entire Key-Value (K-V) heads are removed. As a result, any associated Query heads are also pruned, effectively reducing the dimensionality of the attention mechanism. This is especially relevant in models which employ group-query attention wherein multiple query heads are attended by the same key-value heads thus leading to the pruning of multiple query heads on the removal of a single key-value head. In the case of MLP layers, traditional channel-pruning is adopted for the fully connected layers.

We adopt activation fluctuation based FLAP (An et al., 2023a) which is an approximate metric for structured recoverability, referred to as the *fluctuation metric*. This metric is calculated by taking the sample variance of each input feature and weighting it by the squared norm of the corresponding column in the weight matrix. This approach allows us to estimate the recoverability of specific features in a structured manner.

The chosen pruning methods, ShortGPT for depth pruning and FLAP for width pruning, fairly represent the state-of-the-art approaches in the literature for training-free structured pruning. ShortGPT captures the essence of depth pruning by leveraging a straightforward yet effective activation similarity metric, which aligns with the broader class of techniques that aim to identify and eliminate redundant layers based on their contribution to the overall model representation. Similarly, FLAP embodies the principles of width pruning by assessing the structured recoverability of channels and heads, which is consistent with other methods that focus on pruning individual components within a layer. To ensure that FLAP adequately represents the state-of-the-art in width pruning, we compare it against LLM-Pruner (Ma et al., 2023), a competitive method in this domain, across our benchmark suite. The results, presented in the Appendix 6.5, consistently demonstrate the superiority of FLAP over LLM-Pruner, thereby supporting our choice of width pruning technique. By selecting these representative methods, we ensure that our evaluation encompasses the primary paradigms of structured pruning while remaining computationally feasible for large-scale models.

### 3.3 Extended Evaluation

We present an extensive empirical analysis across multiple LLM application areas. Specifically, we focus on the primary common sense reasoning tasks reported in existing works but we extend them to more niche and practical application areas such as mathematics and coding ability. Additionally, we also focus on instruction following ability of pruned LLMs and the influence of in-context prompts in model predictions. We also investigate the impact of pruning on multimodal understanding of LLMs. Finally, we present a thorough investigation on the bias and fairness of pruned LLMs. We compare all the above mentioned tasks across depth and width pruning regimes and multiple compression ratios. This extended evaluation provides insights into real world performance of pruned LLMs. Finally, we provide insights on the appropriate task-specific pruning methodology.

# 4 Empirical Analysis

## 4.1 Choice of Datasets

This section outlines datasets for evaluating pruned LLMs across reasoning, language modeling, code generation, instruction following, fairness, and multimodal tasks. Key benchmarks include ARC, GSM8k, MMLU, Wikitext-2, HumanEval, IFEVAL, StereoSet, POPE, TextVQA, and MMMU. These datasets assess linguistic, logical, and multimodal capabilities comprehensively. The details of each of the mentioned tasks are provided in the Appendix.

## 4.2 Choice of LLMs

We selected a diverse range of large language models for pruning and evaluation on reasoning-based tasks, including LLaMA-3-8B (Dubey et al., 2024), Gemma-2-9B, Gemma-2-2B (Team et al., 2024), and Qwen-2-1.5B (Yang et al., 2024). These models were chosen for their mass adoption in real-world applications and parameter scale where pruning matter much more as compared to >50B parameters. By applying pruning techniques, we assess each model's efficiency in terms of throughput and their performance on a diverse selection of tasks. For tasks requiring instruction following, we use instruction-tuned versions of LLaMA-3-8B and Gemma-2-9B. These instruction-tuned models are optimized to better understand and follow task-specific instructions.

## 4.3 Incompleteness of current evaluation schemes for LLM pruning

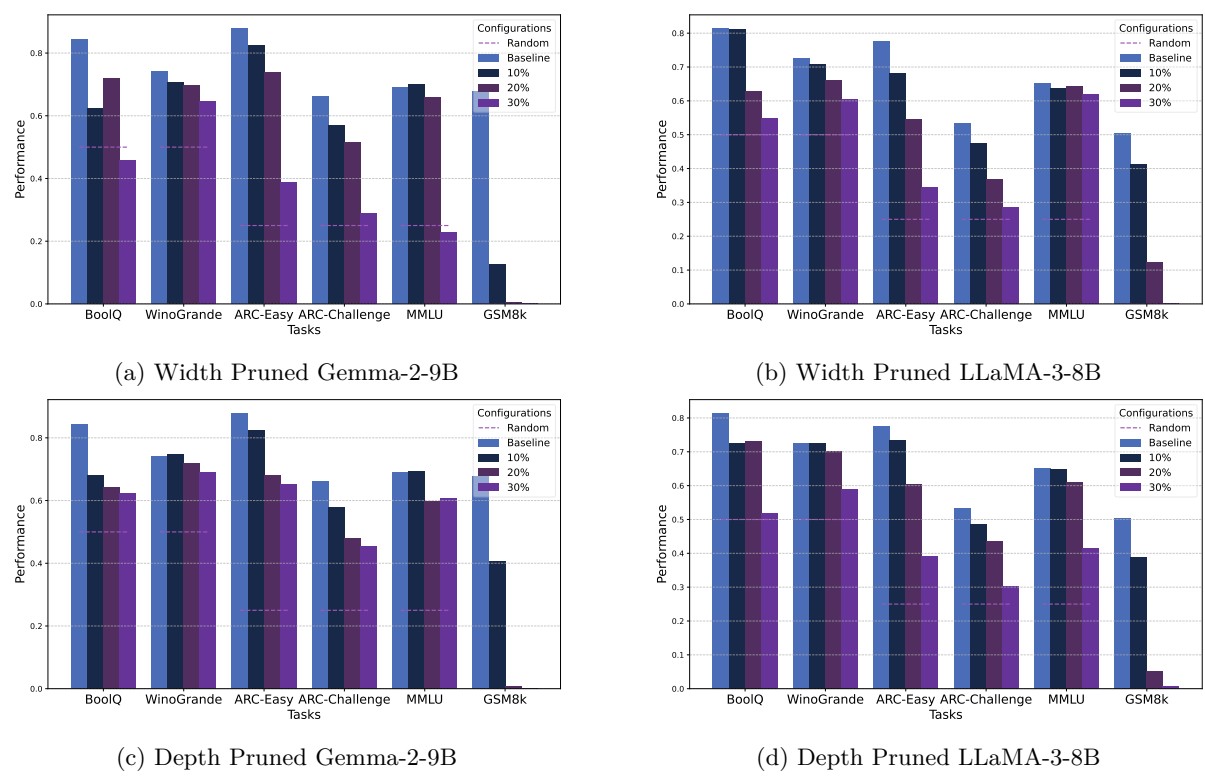

(a) Width Pruned Gemma-2-9B

(b) Width Pruned LLaMA-3-8B

(c) Depth Pruned Gemma-2-9B

(d) Depth Pruned LLaMA-3-8B

Figure 1: Evaluations on Extended Benchmarks for Width and Depth Pruned Models : The figure compares the performance of width-pruned (1a, 1b) and depth-pruned (1c, 1d) versions of Gemma-2-9B and LLaMA-3-8B across varying compression ratios (10%, 20%, and 30%) on extended reasoning benchmarks: BoolQ, WinoGrande, ARC-Easy, ARC-Challenge, MMLU, and GSM8K. Baseline performances and random thresholds are included for reference.

This section highlights the impact of depth and width pruning on numerous downstream tasks across varying budgets. BoolQ, ARC-Challenge, ARC-Easy, and WinoGrande are widely recognized common sense reasoning benchmarks, frequently reported in the majority of pruning literature as performance metrics. In addition to these, we propose evaluating models on MMLU and GSM8K, which are knowledge-based reasoning tasks. MMLU serves as a benchmark for assessing the knowledge acquired by language models during pre-training, covering 57 categories and making it a knowledge-intensive task. In contrast, GSM8K is designed to evaluate high-school-level mathematical reasoning skills. While foundational models frequently report performance on these benchmarks, we argue that it is equally important to include these evaluations in pruning literature. Doing so provides a more comprehensive understanding of the capabilities of compressed models. Papers introducing foundational models Dubey et al. (2024); Yang et al. (2024) typically present a comprehensive suite of benchmarks to demonstrate the model's versatility across diverse tasks. In contrast, pruning literature often focuses on a limited set of simpler benchmarks, primarily in the domain of common sense reasoning. In rare instances, datasets like MMLU are included, highlighting a narrower scope of evaluation. Our findings in Figure 1 support Siddiqui et al. (2024), which shows significant performance drops in depth-pruned models on GSM8K even at low compression ratios. Extending this trend, we find that width-pruned models exhibit a similar decline. The complete evaluations of Qwen-2-1.5B and Gemma-2-2B across varying budgets using both depth and width pruning are provided in the Appendix.

### 4.4 Evaluation beyond the norm

**Coding Ability**

We evaluate the code completion ability using the HumanEval dataset, focusing on functional correctness. For each sample in the test set, we generate five completions with a temperature setting of 0.8 and compute the pass@1 metric across the entire dataset to assess performance. We conduct evaluations on the Gemma-2-9B and LLaMA-3-8B models, compressed by 10% and 20% using both depth and width pruning methods. Our findings as shown in Figure 2 indicate that pruning has a profoundly negative impact on the code completion abilities of models. Regardless of the pruning method employed, even a modest compression ratio of 10% results in a significant decline in pass@1 performance. This pattern is consistent across both the Gemma-2-9B and LLaMA-3-8B model series. Furthermore, we observe that both depth and width pruning methods have a similarly detrimental effect on the models' code completion capabilities. At higher compression ratios, the models lose their ability to maintain proper syntax and begin generating outputs with random, repetitive tokens, further highlighting the adverse effects of aggressive pruning on their functionality. These findings underscore that existing evaluations of pruned LLMs often overstate their effectiveness due to incomplete or limited evaluation tasks, necessitating a critical reassessment of the true value of pruning.

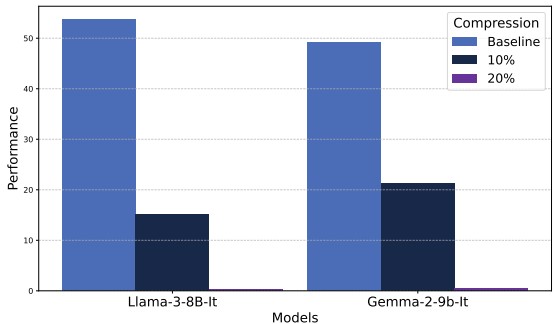
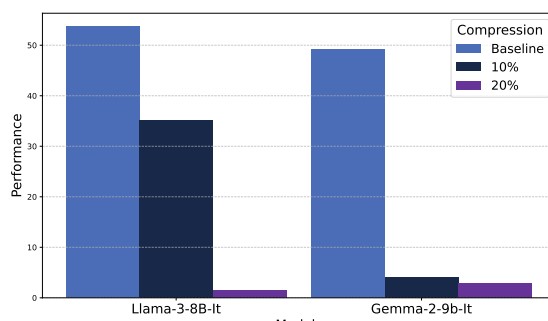

(a) Pass@1 performance of Depth Pruned Models  (b) Pass@1 performance of Width Pruned Models

Figure 2: The figure illustrates the Pass@1 performance of depth-pruned 2a and width-pruned 2b versions of LLaMA-3-8B and Gemma-2-9B models on Human-Eval at different compression ratios (10% and 20%), alongside their respective baselines.

**Instruction Following**

The evaluations in Table 1 reveals that the performance degradation becomes significant as compression ratios increase, with width pruning generally leading to sharper declines compared to depth pruning. While depth pruning retains better performance at lower compression levels, the gap narrows at higher ratios, where both strategies exhibit severe losses. This benchmark is often overlooked in pruning literature, as our evaluations reveal a far greater loss in performance compared to the more commonly reported metrics.

Table 1: IFEVAL scores for LLaMA-3-8B and Gemma-2-2B Instruction Tuned Model computed across varying budgets (10%, 20% and 30% specifically for LLaMA-3-8B)

| Model | Strategy | Compression (%) | Score |
|---|---|---|---|
| **LLaMA-3-8B** | Baseline | - | 0.410 |
| | Depth | 10 | 0.356 |
| | | 20 | 0.216 |
| | | 30 | 0.130 |
| | Width | 10 | 0.301 |
| | | 20 | 0.258 |
| | | 30 | 0.110 |
| **Gemma-2-2B** | Baseline | - | 0.321 |
| | Depth | 10 | 0.255 |
| | | 20 | 0.144 |
| | Width | 10 | 0.125 |
| | | 20 | 0.097 |

Table 2: Depth and width pruned LLaMA-3-70B results on 20% and 40% compression on reasoning and Wikitext

| Method | Compression (%) | ARC_E | ARC_C | Winogrande | WikiText | BoolQ | MMLU |
|---|---|---|---|---|---|---|---|
| Baseline | 0 | 86.91 | 64.33 | 80.58 | 2.91 | 85.17 | 75.02 |
| Depth | 20 | 80.85 | 57.25 | 79.40 | 9.02 | 85.56 | 74.65 |
| | 40 | 58.04 | 42.15 | 76.08 | >1e6 | 66.85 | 68.07 |
| Width | 20 | 32.42 | 42.13 | 65.35 | 2.9e5 | 84.34 | 62.99 |
| | 40 | 31.65 | 29.60 | 65.75 | >1e6 | 46.36 | 72.08 |

## 4.5 Depth vs. Width pruning of LLMs

We now investigate how depth and width pruning affects model performance, focusing on their unique impact across various tasks. Based on our evaluations presented in Figure 3 we find that depth pruning performs better overall compared to width pruning. Depth pruning, while reducing the number of layers, appears to preserve the model's ability to extract meaningful hierarchical features, which are crucial for tasks involving logical reasoning and factual knowledge. ARC-Easy, ARC-Challenge and BoolQ consistently benefit from depth pruning as compared to width pruning while on the other hand MMLU and GSM8k show mixed results depending on the model and compression level. When we run standard evaluations on Llama-3-70B for reasoning tasks, we observe a similar trend where depth pruned models consistently outperform those that are width pruned on average. This observation, as detailed in Table 2, reinforces our claim that depth pruning better preserves the model's critical reasoning capabilities while achieving compression. We find that the Gemma-2 series of models is particularly susceptible to significant performance deterioration introduced by width pruning. As observed in our evaluations in Figure 3a and Figure 3c, the degradation becomes more pronounced as the pruning ratio increases, with tasks like ARC-Easy, ARC-Challenge, and MMLU showing clear advantages for depth pruning over width pruning.

However, we also find that perplexity reflects higher degradation upon depth pruning compared to width pruning as shown in Figure 4. This trend indicates that depth pruning, while preserving performance

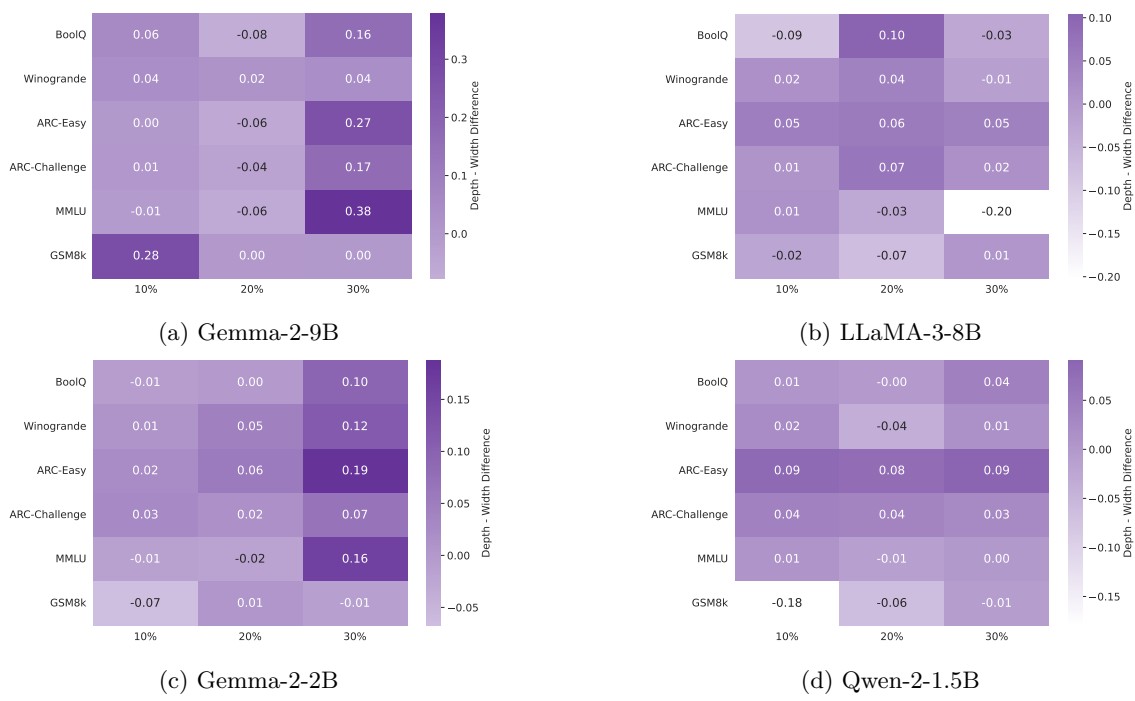

(a) Gemma-2-9B          (b) LLaMA-3-8B

(c) Gemma-2-2B          (d) Qwen-2-1.5B

Figure 3: Difference in Performance of Depth and Width Pruned Models :The figure showcases the performance difference between depth-pruned and width-pruned models (Depth - Width) across varying compression ratios (10%, 20%, and 30%) on extended reasoning benchmarks.

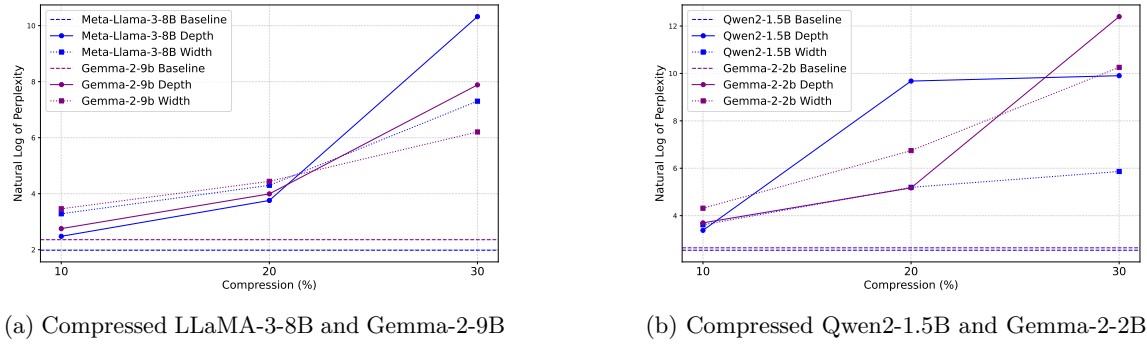

(a) Compressed LLaMA-3-8B and Gemma-2-9B      (b) Compressed Qwen2-1.5B and Gemma-2-2B

Figure 4: Perplexity Performance on WikiText-2 for Compressed Models : The figure compares the natural log of perplexity for depth- and width-pruned models at compression ratios of 10%, 20%, and 30% on the WikiText-2 dataset. Subfigure (4a) shows the results for LLaMA-3-8B and Gemma-2-9B, while subfigure (4b) presents those for Qwen2-1.5B and Gemma-2-2B. The baseline perplexity for each model is represented by dashed lines.

on certain reasoning tasks, significantly disrupts the model's ability to generalize across broader language modeling objectives. At higher pruning ratios, the difference in perplexity between depth-pruned and width-pruned models becomes substantial, often reaching an order or even multiple orders of magnitude higher. This highlights a critical trade-off: while depth pruning retains task-specific performance more effectively, it imposes a severe penalty on perplexity, suggesting a greater loss of overall language modeling capacity compared to width pruning.

### 4.6 Throughput Measurement of Depth and Width Pruned Models

To evaluate the efficiency of depth and width pruned models across different pruning budgets, we measure throughput in terms of tokens generated per second with a batch size of 1. This setup as shown in Table 3 allows us to observe the effect of varying pruning strategies on single-token generation speed. Depth pruning and width pruning achieve computational reductions in fundamentally different ways. Depth pruning focuses on removing entire transformer blocks, significantly decreasing both memory usage and inference latency, as it directly reduces the number of sequential operations in the forward pass. In contrast, width pruning targets the model's internal structures, such as reducing the number of attention heads or the size of MLP layers within transformer blocks. While width pruning reduces the per-layer computation cost and memory requirements, it does not reduce the depth of the model, meaning the number of sequential layers remain constant. As a result, depth pruning offers more pronounced reductions in latency as compared to width pruning.

Table 3: Throughput in tokens/second for LLaMA-3-8B and Gemma-2-2B. The table reports the throughput (measured in tokens per second) for LLaMA-3-8B and Gemma-2-2B models under different pruning strategies (Depth and Width) and compression ratios (10%, 20%, and 30%). The baseline throughput without pruning is included as a reference.

| Model | Pruning Strategy | Compression % | Value (Tokens/Second) |
|---|---|---|---|
| **LLaMA-3-8B** | Baseline | - | 37.16 |
| | Depth | 10% | 41.90 |
| | | 20% | 45.72 |
| | | 30% | 54.40 |
| | Width | 10% | 38.26 |
| | | 20% | 37.98 |
| | | 30% | 39.12 |
| **Gemma-2-2B** | Baseline | - | 31.35 |
| | Depth | 10% | 31.95 |
| | | 20% | 35.87 |
| | | 30% | 44.54 |
| | Width | 10% | 28.65 |
| | | 20% | 28.48 |
| | | 30% | 28.66 |

### 4.7 Effect of Pruning on Bias and Fairness

For a holistic evaluation scheme, we evaluate stereotypical biases in LLaMA-3-8B ( Figure 5 ) and Gemma-2-9B models ( Figure 7 ), both depth and width pruned, by reporting the ICAT scores on the StereoSet dataset. The evaluation includes bias measurements across categories such as race, religion, gender, and profession, along with the computation of the overall ICAT score for each model. It has been noted that biases tend to decrease as language models undergo higher compression Ramesh et al. (2023). Through our experiments, we find that the overall stereotypical bias remains intact consistently across budgets and models. Interestingly, in certain individual categories such as race, we observe an improvement in ICAT scores when compared to the baseline after pruning, indicating a reduced stereotypical bias post-pruning.

### 4.8 Effect of pruning on multimodal ability

Vision-Language Models (VLMs) integrate visual and textual modalities to perform tasks requiring cross-modal understanding, such as image captioning and visual question answering. To evaluate the effect of depth and width pruning on the multimodal performance of VLMs, we first isolate the language model of Idefics3-8B (Laurençon et al., 2024), which is a fine-tuned LLaMA3-8B model, and prune it under various budgets. The pruned language model is then reconnected to the vision encoder, and the full model's performance is assessed across multiple multimodal benchmarks as shown in Table 4. Through this experiment we find that depth

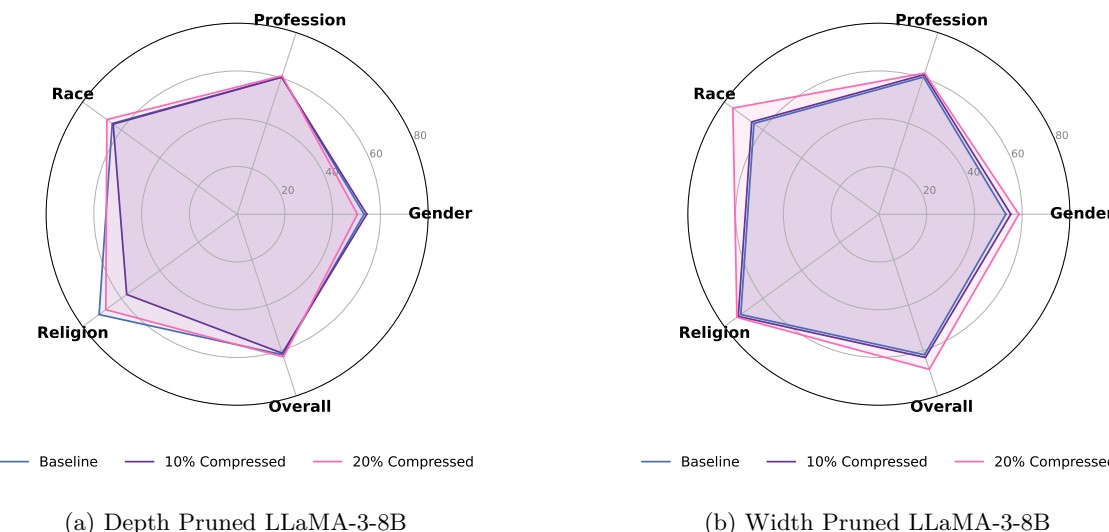

(a) Depth Pruned LLaMA-3-8B                    (b) Width Pruned LLaMA-3-8B

Figure 5: ICAT Scores for Depth and Width Pruned LLaMA-3-8B. The figure presents the ICAT scores across five domains—Gender, Profession, Race, Religion, and Overall—for depth-pruned (5a) and width-pruned (5b) versions of LLaMA-3-8B at varying compression levels (10% and 20%) and the baseline.

pruning performs better overall which aligns with our previous findings. Moreover, we observe that while performing width pruning, as we increase the compression ratio POPE score decreases drastically suggesting that the model begins to hallucinate although it's performance on other benchmarks is retained comparatively. However, during depth pruning the model loses its Text Understanding capabilities as the pruning ratio increases.

Table 4: The table presents the performance of Idefics3-8B across three benchmarks—POPE, TextVQA, and MMMU—under varying pruning strategies (Depth and Width) and compression ratios (10% and 20%).

| Pruning Strategy | Compression % | POPE | TextVQA | MMMU |
|---|---|---|---|---|
| Baseline | - | 86.72 | 60.42 | 43.33 |
| Depth | 10% | 87.17 | 54.66 | 41.56 |
|  | 20% | 72.71 | 25.38 | 38.44 |
| Width | 10% | 86.99 | 50.16 | 42.56 |
|  | 20% | 54.24 | 37.20 | 39.11 |

## 4.9 Effect of pruning on long-context handling ability

We present evaluations on long-context benchmarks using the LLaMA3-8B-Instruct model within the LEval framework (An et al., 2023b) in Table 5. LEval includes multiple challenging tasks: QuALITY, which consists of questions based on long literary stories from Gutenberg; Coursera, designed to assess the reasoning ability of LLMs on complex and difficult-to-comprehend course materials; SFiction, which features questions that contradict real-world principles, testing the model's ability to rely on input context rather than pretrained knowledge; and GSM100, a subset of 100 grade school math problems from the GSM8K dataset, aimed at evaluating few-shot learning capabilities in very long-context scenarios. The table below presents the performance of different pruning strategies at 10% and 20% compression on long-context benchmarks, compared to the Baseline. Depth pruning initially improves performance on SFiction but degrades on Coursera and GSM100 as pruning increases. Width pruning shows a trade-off, with 10% pruning slightly improving overall performance but 20% pruning significantly reducing accuracy, especially on GSM100.

Overall in the case of long context understanding, we do not see a clear pattern with respect to performance degradation.

Table 5: Comparison of pruning strategies at different compression levels across long-context evaluation benchmarks

| Strategy | Pruning (%) | Quality | Coursera | SFcition | GSM100 |
|---|---|---|---|---|---|
| Baseline | - | 62.38 | 52.33 | 69.53 | 80.00 |
| Depth | 10% | 62.38 | 47.24 | 75.00 | 74.00 |
| | 20% | 62.87 | 42.88 | 67.19 | 39.00 |
| Width | 10% | 63.86 | 44.19 | 70.31 | 65.00 |
| | 20% | 60.40 | 51.45 | 67.97 | 39.00 |

### 4.10 Choosing the right pruning strategy

Depth pruning has been shown to outperform width pruning at moderate compression ratios, particularly for reasoning-intensive tasks such as GSM8K and knowledge benchmarks like MMLU. However, perplexity evaluations demonstrate that depth pruning can significantly degrade language modeling capabilities. The removal of entire layers disrupts the model's depth-driven representation learning, which is essential for capturing sequential dependencies in language tasks.

In contrast, width pruning achieves compression by reducing the dimensionality of individual layers. This method is more effective for maintaining language modeling performance, as it preserves the model's depth. However, at higher compression levels, width pruning disproportionately reduces the model's representational capacity, leading to a sharper decline in downstream task performance, particularly for reasoning and knowledge-intensive tasks.

Depth pruning is therefore more suitable for scenarios that involve reasoning-intensive tasks, require moderate compression ratios of 10–20%, and prioritize significant inference speedups. It is less effective when language modeling capabilities are central to task performance. Conversely, width pruning is more appropriate to preserve language modeling capabilities with minimal performance loss.

In cases where both efficiency and hierarchical depth are critical, combining depth and width pruning in a hybrid approach can help balance compression performance and retention of downstream task accuracy. By strategically leveraging the strengths of both methods, it is possible to mitigate the trade-offs associated with either pruning strategy.

### 4.11 Fairing Against Unstructured Pruning on Extended Benchmarks

While we focus on evaluating structured pruning methods, it is important to understand how these fair against unstructured pruning methods such as Wanda and SparseGPT on our extended benchmarks. Our experiments with SparseGPT (Frantar and Alistarh, 2023) and Wanda (Sun et al., 2024) on LLaMA3-8B, Table 6, span common sense reasoning tasks, mathematical reasoning (GSM8K), Code Eval, and Instruction Following (IFEVAL). Our findings suggest that unstructured and semi-structured pruning methods, which allow for finer granularity, generally preserve downstream task performance better than structured pruning approaches that reduce model depth or width. The only task where these methods exhibit significant performance degradation is mathematical reasoning. These results offer further insight into the impact of unstructured pruning on LLMs and highlight that, when the target hardware supports unstructured sparsity, these methods may be preferable over structured pruning techniques.

## 5 Conclusions

Training-free structured pruning of LLMs presents significant challenges, as evidenced by our comprehensive evaluations. Our results demonstrate that such pruning methods can lead to substantial performance

Table 6: Results for unstructured pruning methods on extended benchmarks with different compression ratios.

| Pruning Method | Compression (%) | ARC_C | ARC_E | Winogrande | BoolQ | MMLU | GSM8K | CodeEval | WikiText | IFEVAL |
|---|---|---|---|---|---|---|---|---|---|---|
| Wanda | 10% | 79.88 | 50.42 | 73.40 | 81.47 | 61.81 | 50.03 | 48.11 | 5.58 | 40.29 |
| | 20% | 79.71 | 49.14 | 72.61 | 81.87 | 61.11 | 47.83 | 49.90 | 5.71 | 39.18 |
| | 30% | 78.57 | 48.38 | 73.40 | 81.80 | 59.31 | 41.39 | 48.11 | 6.03 | 41.03 |
| SparseGPT | 10% | 80.38 | 50.34 | 72.92 | 81.71 | 61.69 | 47.91 | 49.14 | 5.58 | 41.22 |
| | 20% | 79.59 | 49.31 | 72.85 | 81.68 | 60.95 | 48.06 | 48.28 | 5.72 | 41.77 |
| | 30% | 78.70 | 47.61 | 74.11 | 81.89 | 59.88 | 43.59 | 47.93 | 6.05 | 39.74 |

degradation, particularly on benchmarks that are often overlooked. Highlighting these underperforming benchmarks is essential, as it encourages the development of new methods that address these shortcomings and improve overall model robustness.

We have showcased the differences between depth and width pruning strategies to gain a deeper understanding of their disparate impacts on downstream tasks. This disparity emphasizes the need for careful selection of pruning strategies based on the specific requirements of the task at hand.

By bringing attention to these challenges and disparities, we hope this paper inspires future research in the compression literature to evaluate their methods across a wider variety of tasks. Comprehensive evaluations will not only illuminate the limitations of current approaches but also guide the development of more effective pruning techniques that maintain performance while achieving desired levels of model compression.

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

## 6 Appendix

### 6.1 Details on ShortGPT

ShortGPT calculates the cosine similarity between the input and output embeddings of the $i$th block, using cosine similarity as a global pruning criterion across model layers.

$$B_i = 1 - \frac{\sum_{j=1}^{N} \left( X_{i-1}^{(j)} \cdot X_i^{(j)} \right)}{N} \tag{1}$$

In expression 1, $B_i$ represents the pruning criterion for the $i$-th block, $X_i$ denotes the output embedding of the $i$-th layer, $X_{i-1}$ refers to the output embedding of the $(i-1)$-th layer (serving as the input to the $i$-th layer), $N$ is the total number of samples used in calculating cosine similarity, and $\sum_{j=1}^{N} \left( X_{i-1}^{(j)} \cdot X_i^{(j)} \right)$ represents the summation over all samples, with $X_{i-1}^{(j)}$ and $X_i^{(j)}$ being the embeddings from the $(i-1)$-th and $i$-th layers for the $j$-th sample. After calculating the $B_i$ for each block, we sort them in an ascending order based on the $B_i$ values and prune $k$ blocks with the lowest $B_i$. $k$ is decided depending on the target pruning ratio.

### 6.2 Details on FLAP

Fluctuation metric $S_{,:,j}$ for the $j$th channel of the $l$th layer is defined as :

$$S_{l,j} = \frac{1}{N-1} \sum_{i=1}^{N} (X_{l,j}^{(i)} - \bar{X}_{l,j})^2 \cdot \|W_{l,j}\|^2 \tag{2}$$

Here, $N$ denotes the total number of calibration samples, and $X_{l,j}^{(i)}$ is the value of the $j$th channel of the input features at layer $l$ for the $i$th sample. The term $\bar{X}_{l,j}$ represents the mean of $X_{l,j}$ across all $N$ samples, allowing us to calculate the variance of this feature. Additionally, $\|W_{l,j}\|^2$ refers to the squared norm of the $j$-th column of the weight matrix in layer $l$, which weights the variance term in the metric calculation. Together, these components form an estimate of each channel's recoverability by quantifying its variance and weighting it according to the model's structure.

To ensure that the score can serve as a global approximation, the metric is standardized for each layer to a common mean and standard deviation.

$$S'_{l,j} = \frac{S_{l,j} - \mathbb{E}[S_{l,j}]}{\sqrt{\mathbb{E}[(S_{l,j} - \mathbb{E}[S_{l,j}])^2]}} \tag{3}$$

where $\mathbb{E}[S_{l,j}]$ denotes the expected value of the vector $S_{l,j}$, and $\sqrt{\mathbb{E}[(S_{l,j} - \mathbb{E}[S_{l,j}])^2]}$ represents the standard deviation, calculated as the square root of the variance.

### 6.3 Datasets

In this section, we describe the datasets used as part of our extended evaluation of pruned LLMs:

**Common Sense Reasoning and Factual Question Answering :**

- **ARC Challenge and ARC Easy (Clark et al., 2018):** These are standard benchmarks for common sense reasoning and are widely referenced in pruning literature, providing a basis for comparison with existing models.

- **BoolQ (Clark et al., 2019) and Winogrande (Sakaguchi et al., 2020):** Both datasets are used for binary question-answering tasks that involve common sense and linguistic reasoning, making them essential for understanding how pruning impacts nuanced reasoning abilities.

- **GSM8k (Cobbe et al., 2021):** A dataset designed to test mathematical and logical reasoning skills. This dataset's complexity surpasses that of traditional common sense tasks, offering insight into how pruning affects tasks with higher-order reasoning requirements.

- **MMLU (Hendrycks et al., 2021):** Used alongside GSM8k to evaluate factual knowledge combined with reasoning, allowing us to examine pruning performance on tasks requiring an integration of knowledge domains.

**Wikitext-2 (Perplexity):** Wikitext-2 serves as a standard language modeling benchmark for measuring model perplexity, providing insights into the fluency and coherence of pruned models. Perplexity is a critical metric here, as it quantifies the model's ability to predict word sequences effectively, highlighting any degradation in linguistic capability due to pruning.

**Code Completion on HumanEval (Chen et al., 2021):** The HumanEval dataset is a benchmark for evaluating the functional correctness of code generated by AI models. The primary purpose of the dataset is to test the ability of models to generate syntactically correct and semantically meaningful code that passes all provided unit tests, thus HumanEval emphasizes correctness over superficial metrics like BLEU or code similarity.

**Instruction Following on IFEVAL (Zhou et al., 2023):** The dataset comprises prompts utilized in the Instruction-Following Evaluation (IFEVAL) benchmark, specifically designed for evaluating the performance of large language models. It includes approximately 500 "verifiable instructions", which are carefully constructed tasks that allow for objective and heuristic-based verification.

**Fairness Evaluation on Stereoset (Nadeem et al., 2021):** StereoSet is a benchmark designed to evaluate the social bias present in language models across dimensions like gender, race, religion, and profession. It tests a model's tendency to exhibit stereotypical associations in natural language understanding tasks. A combined metric, the Idealized Context Association Test (ICAT) Score, rewards models that exhibit low bias while maintaining high language quality.

**Multimodal Evaluation:** To evaluate the multimodal capabilities of Vision-Language Models (VLMs), we use three datasets: POPE (Li et al., 2023) for detecting object hallucinations; TextVQA (Singh et al., 2019) for assessing the ability to read and reason about text in images; and MMMU (Yue et al., 2024), which tests college-level subject knowledge and deliberate reasoning covering six core disciplines - Art & Design, Business, Science, Health & Medicine Humanities & Social Science and Tech & Engineering.

### 6.4 Analysis of Pruned Models Under Varying Budgets

Here, we present the complete evaluations of Qwen2-1.5B and Gemma-2-2B (Figure 6) to assess the effects of depth and width pruning on smaller-sized models. The results reveal a consistent trend of performance degradation, particularly on the GSM8K benchmark, even at low compression ratios.

### 6.5 Comparing FLAP with LLM-Pruner

To establish FLAP as a state-of-the-art width pruning method, we compare it against LLMPruner (Ma et al., 2023) across our benchmark suite. Table 7 presents a comparison of the zero-shot performance of LLaMA-7B pruned using LLM-Pruner and FLAP at 20% and 50% pruning ratios, consistently demonstrating FLAP's superiority across tasks.

Table 7: Comparison of methods at different compression levels across multiple benchmarks.

| Method | Compression (%) | ARC_C | ARC_E | BoolQ | Winogrande |
|---|---|---|---|---|---|
| LLM-Pruner | 20% | 37.18 | 59.18 | 59.39 | 61.33 |
| FLAP | 20% | 39.25 | 69.91 | 69.63 | 68.85 |
| LLM-Pruner | 50% | 25.51 | 32.83 | 52.57 | 49.01 |
| FLAP | 50% | 29.95 | 49.66 | 60.21 | 57.54 |

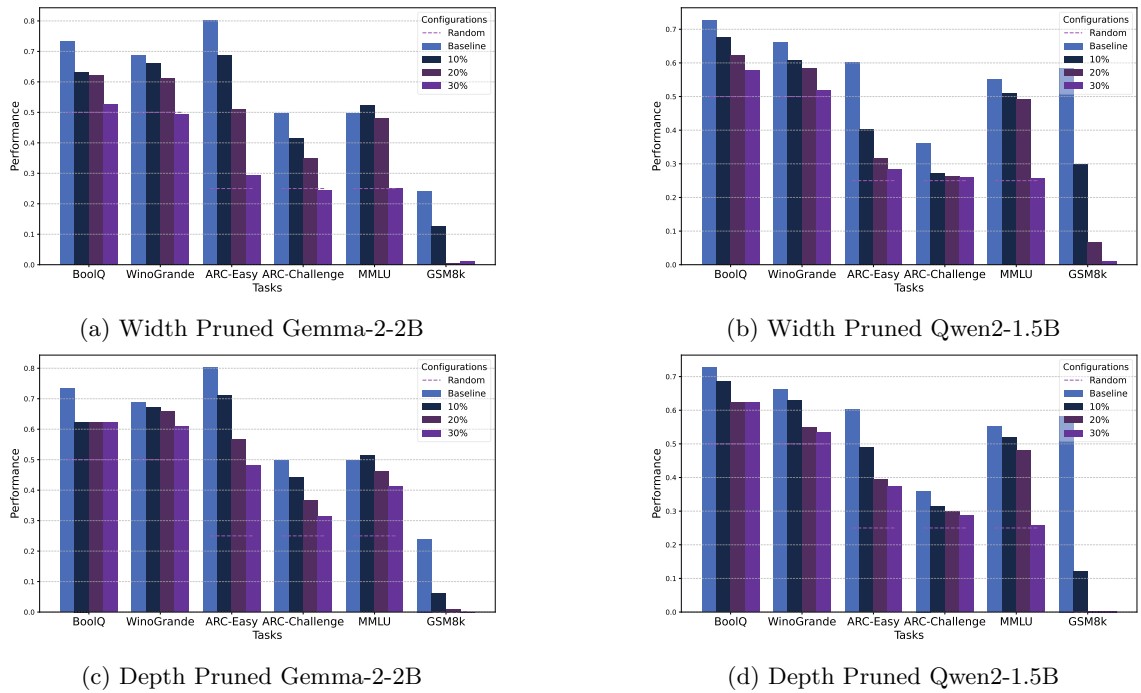

(a) Width Pruned Gemma-2-2B      (b) Width Pruned Qwen2-1.5B

(c) Depth Pruned Gemma-2-2B      (d) Depth Pruned Qwen2-1.5B

Figure 6: Performance of Smaller Models on Extended Benchmarks : The figure presents the performance of Gemma-2-2B (6c,6a) and Qwen2-1.5B (6d,6b) on extended reasoning benchmarks, comparing depth and width pruning

## 6.6 Bias and Fairness Evaluation in Gemma-2-9B

We evaluate Gemma-2-9B model (Figure 7) on the StereoSet dataset to analyze its behavior regarding stereotypical biases under the influence of depth and width compression across varying budgets.

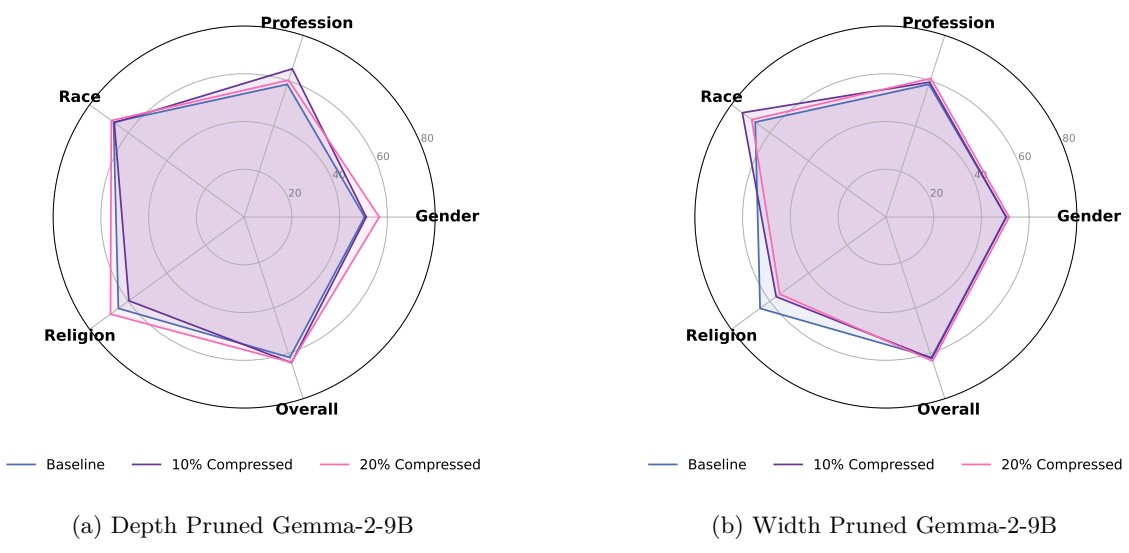

(a) Depth Pruned Gemma-2-9B      (b) Width Pruned Gemma-2-9B

Figure 7: Bias Evaluations for Gemma-2-9B : The radar plots illustrate the ICAT scores across domains—Gender, Profession, Race, Religion, and Overall—for depth-pruned (7a) and width-pruned (7b) Gemma-2-9B

