# OpenReview forum: "Rethinking the Value of Training-Free Structured Pruning of LLMs"
_TMLR — Accepted by TMLR_

### Review · Reviewer_EncY · 2025-02-04

**Summary Of Contributions:**

The paper evaluates training-free structured pruning techniques for Large Language Models (LLMs), comparing depth and width pruning strategies. It finds that while some tasks handle pruning well, others degrade significantly, challenging previous studies that used selective benchmarks. The paper shows that depth pruning consistently works better than width pruning, highlighting the need for better pruning methods and more comprehensive evaluations.

**Audience:**

Yes

**Broader Impact Concerns:**

I do not have concerns regarding broader impacts of this paper.

**Claims And Evidence:**

Yes

**Requested Changes:**

See weaknesses above.

**Strengths And Weaknesses:**

- “The chosen pruning methods, ShortGPT for depth pruning and FLAP for width pruning, fairly represent the state-of-the-art approaches in the literature for training-free structured pruning.”

    Could the authors provide some supporting evidence for this? For example, some reference works that discuss ShortGPT and FLAP.

    Similarly, claims like this should be supported with concrete evidence:
    “By selecting these representative methods, we ensure that our evaluation encompasses the primary paradigms of structured pruning while remaining computationally feasible for large-scale models.”
    “In contrast, pruning literature often focuses on a limited set of simpler benchmarks, primarily in the domain of common sense reasoning”

- Smaller models may be prone to accuracy degradation during pruning. It would be good to results on models with sizes larger than 8b, e.g., llama-3.3-70b?

- The results are interesting, showing that pruning can lead to catastropic drop of performance in certain task, e.g., coding. However, it would be good to add some analysis or discussions on this observation.

- I am not sure about the focus of “Training-free” in the paper. In principle, pruning is a technique that often involves fine-tuning [1].

- It would be also to see if similar observations can be found in sparse pruning methods [2,3].

[1] Rethinking the value of network pruning. ICLR, 2019.
[2] SparseGPT: Massive Language Models Can Be Accurately Pruned in One-Shot. ICML 2023.
[3] A simple and effective pruning approach for LLMs. ICLR 2024.

---

> ### Author Response · Authors · 2025-02-27
> **Official Comment 1/3**
>
> We appreciate the reviewer for highlighting our observation that existing training-free pruning methods show results on selective benchmarks.
>
> ### W1: “The chosen pruning methods, ShortGPT for depth pruning and FLAP for width pruning, fairly represent the state-of-the-art approaches in the literature for training-free structured pruning.” Could the authors provide some supporting evidence for this? For example, some reference works that discuss ShortGPT and FLAP. Similarly, claims like this should be supported with concrete evidence: “By selecting these representative methods, we ensure that our evaluation encompasses the primary paradigms of structured pruning while remaining computationally feasible for large-scale models.” “In contrast, pruning literature often focuses on a limited set of simpler benchmarks, primarily in the domain of common sense reasoning”
>
> For depth pruning, we cite [1], [2], and [3], as these recent methods achieve results competitive with training-free depth pruning SOTA. They utilize cosine similarity to quantify changes in model representations across blocks and layers, employing it as a measure of importance and thereby establishing it as a representative approach for this class of structured pruning. Additionally [4] discusses training-free structured pruning methods in detail and mentions LLM-Pruner and FLAP as strong width pruning baselines. All of these methods are computationally feasible for large scale models able to prune on a single 40GB GPU for <8B parametric model and cover the primary paradigms - depth and width of structured pruning.
>
> Additionally all the aforementioned papers only evaluate on perplexity and a few common sense reasoning benchmarks[1,2,3,4], solidifying our claim on the need of an extended evaluation. We have updated our final draft to reflect these changes.
>
>
> ### W2: Smaller models may be prone to accuracy degradation during pruning. It would be good to results on models with sizes larger than 8b, e.g., llama-3.3-70b?
>
> We appreciate the reviewer’s feedback on evaluating larger models. In response, we have conducted additional experiments on LLaMA-3-70B. We have included these results in the revised version to further support our findings, highlighting the superiority of depth pruning in most cases. An interesting thing to note is that even though the wikitext perplexity rises exponentially, performance of the width pruned model on downstream tasks is comparatively intact. Highlighting that perplexity is not an ideal measure to gauge pruned model performance.
>
> **Evaluations on LLaMA-3-70B**
>
> | Method   | Pruning (%) | ARC_E | ARC_C | Winogrande | WikiText | BoolQ  | MMLU  |
> |----------|------------|-------|-------|------------|----------|--------|-------|
> | Baseline | 0          | 86.91 | 64.33 | 80.58      | 2.91     | 85.17  | 75.02 |
> | Depth    | 20         | 80.85 | 57.25 | 79.40      | 9.02     | 85.56  | 74.65 |
> | Depth    | 40         | 58.04 | 42.15 | 76.08      | >1e6     | 66.85  | 68.07 |
> | Width    | 20         | 32.42 | 42.13 | 65.35      | 2.9e5    | 84.34  | 62.99 |
> | Width    | 40         | 31.65 | 29.60 | 65.75      | >1e6     | 46.36  | 72.08 |
>
>
> In addition, we will include 70B evaluations on Instruction Following and Code Completion in the final submission.

---

> ### Author Response · Authors · 2025-02-27
> **Official Comment 2/3**
>
> ### W3: The results are interesting, showing that pruning can lead to catastropic drop of performance in certain task, e.g., coding. However, it would be good to add some analysis or discussions on this observation.
>
>
>
> Our evaluations on the HumanEval Code Completion dataset reveal that pruning often causes the model to generate excessively long code completions, frequently due to repetitive token generation. Below are examples illustrating this behavior in a 10% depth- and width-pruned LLaMA-3-8B model. These qualitative results show that in complex scenarios such as code completion, pruned models undergo catastrophic degradation. For example :-
>
> **Sample Input Prompt**
> ```python
> from typing import List
> def  intersperse(numbers: List[int], delimeter: int) -> List[int]:
> """ Insert a number 'delimeter' between every two consecutive elements of input list `numbers'
> >>> intersperse([], 4)
> []
> >>> intersperse([1, 2, 3], 4)
> [1, 4, 2, 4, 3]
> """
> ```
>
> **Code Block A**
> ```python
> from typing import List
> def  intersperse(numbers: List[int], delimeter: int) -> List[int]:
> """ Insert a number 'delimeter' between every two consecutive elements of input list `numbers'
> >>> intersperse([], 4)
> []
> >>> intersperse([1, 2, 3], 4)
> [1, 4, 2, 4, 3]
> """
> if  not numbers:
> return numbers
> result = [numbers[0]]
> for num in numbers[1:]:
> result.extend([num, delimeter])
> return result
> ```
>
> **Code Block B**
> ```python
> from typing import List
> def intersperse(numbers: List[int], delimeter: int) -> List[int]:
> """ Insert a number 'delimeter' between every two consecutive elements of input list `numbers'
> >>> intersperse([], 4)
> []
> >>> intersperse([1, 2, 3], 4)
> [1, 4, 2, 4, 3]
> """
> if  not numbers:
> return []
> if  len(numbers == 1):
> return [a]
> return [numbers[0] if i == 0  else del if i == ]
>
> if  not numbers:
> return []
> if  len(numbers == 1):
> return [a]
> result = []
> result.append(numbers[0]
> for i, number  in enumerate(numbers):
> if < 1:
> result.append(delimeter)
> result.append(number)
> return  result
>
> if  not numbers:
> return []
> if  len(numbers == 1):
> return [a]
> result = []
> result.append(numbers[0]
> for i, number  in enumerate(numbers):
> if < 1:
> result.append(delimiter)
> result.append(number)
> return  result
> ```
>
> **Code Block C**
> ```python
> from typing import List
> def  intersperse(numbers: List[int], delimeter: int) -> List[int]:
> """ Insert a number 'delimeter' between every two consecutive elements of input list `numbers'
> >>> intersperse([], 4)
> []
> >>> intersperse([1, 2, 3], 4)
> [1, 4, 2, 4, 3]
> """
> result: List[int] = []
> for i, x in enumerate(numbers):
> if i!= len(numbers - 1]:
> result.append(x)
> result.append(dleemter)
> isnd-send. and (.
> .............. ...... .. . .. . . . . . .
> ```
>
> Code Blocks A, B, and C present sample completions from the Baseline, 20% Width-Pruned, and Depth-Pruned LLaMA-3-8B models, respectively. Our findings indicate that sparsity leads to significant model degradation in code completion. This degradation commonly manifests in two ways: (1) generating incorrect code, such as using undefined variables or producing repetitive code blocks, as observed in Code Block B, and (2) generating sequences of repeating, non-ending characters, as seen in Code Block C. Based on our analysis of generated completions, these failure modes are the most frequent indicators of performance decline in pruned models when applied to code completion tasks.
>
>
>
>
> ### W4: I am not sure about the focus of “Training-free” in the paper. In principle, pruning is a technique that often involves fine-tuning
>
>
> We acknowledge the reviewer’s concern on the focus of this work being on “Training-free” while most fundamental pruning methods usually have a fine-tuning stage post pruning. Traditional deep learning models like ResNet-50 (24M parameters) and BERT-Base (108M parameters) have significantly fewer parameters compared to LLMs such as LLaMA 3 (8B parameters). This makes full fine-tuning nearly impossible for users without access to massive GPU clusters. To address this challenge, a new class of training-free pruning methods have recently emerged [1,2,3]. These methods bypass the need for a dedicated fine-tuning stage, instead utilizing the pruned architecture as the final model.
>
> One might argue that parameter-efficient fine-tuning (PEFT) techniques, such as LoRA, could serve as an alternative to full fine-tuning. However, PEFT methods make only minimal modifications to the base model’s weights. As a result, better training-free pruned weights directly lead to improved final PEFT performance.
>
> Given these factors, our study aims to **shed light on the specific effects and challenges of training-free structured pruning.** We acknowledge that there exist pruning strategies that integrate fine-tuning post-compression, but our work deliberately focuses on **the domain where retraining is not an option.**

---

> ### Author Response · Authors · 2025-02-27
> **Official Comment (3/3)**
>
> ### W5: It would be also to see if similar observations can be found in sparse pruning methods
>
> Upon the recommendation of the reviewer, we have added experiments with both SparseGPT and Wanda on common sense reasoning benchmarks, mathematical reasoning (GSM8K), Code Eval and Instruction Following (IFEVAL). Our findings indicate that these unstructured methods, which offer finer granularity in pruning, better preserve performance on downstream tasks compared to structured - depth and width pruning approaches. The only benchmark where these methods show substantial deterioration is mathematical reasoning. This experiment provides further insights on how unstructured pruning affects LLMs and if the target hardware supports unstructured sparsity, these methods are a better choice as compared to structured - depth and width pruning approaches.
>
> **Results on SparseGPT and Wanda across varying budgets and downstream tasks.**
> | Pruning Method | Sparsity (%) | ARC_E | ARC_C | Winogrande | BoolQ | MMLU  | GSM8k | CodeEval | WikiText | IFEVAL |
> |---------------|---------|-------|-------|------------|-------|------|-------|---------|---------|--------|
> | Wanda        | 10      | 79.88 | 50.42 | 73.40      | 81.47 | 61.81 | 50.03 | 48.11   | 5.58    | 40.29  |
> | Wanda        | 20      | 79.71 | 49.14 | 72.61      | 81.87 | 61.11 | 47.83 | 49.90   | 5.71    | 39.18  |
> | Wanda        | 30      | 78.57 | 48.38 | 73.40      | 81.80 | 59.31 | 41.39 | 48.11   | 6.03    | 41.03  |
> | SparseGPT    | 10      | 80.38 | 50.34 | 72.92      | 81.71 | 61.69 | 47.91 | 49.14   | 5.58    | 41.22  |
> | SparseGPT    | 20      | 79.59 | 49.31 | 72.85      | 81.68 | 60.95 | 48.06 | 48.28   | 5.72    | 41.77  |
> | SparseGPT    | 30      | 78.70 | 47.61 | 74.11      | 81.89 | 59.88 | 43.59 | 47.93   | 6.05    | 39.74  |
>
>
> ### References
> [1] [https://arxiv.org/pdf/2403.17887](https://arxiv.org/pdf/2403.17887)
>
> [2] [https://arxiv.org/abs/2406.15786](https://arxiv.org/abs/2406.15786)
>
> [3] [https://arxiv.org/pdf/2402.11187](https://arxiv.org/pdf/2402.11187)
>
> [4] [https://aclanthology.org/2024.emnlp-main.775.pdf](https://aclanthology.org/2024.emnlp-main.775.pdf)

---

> ### Comment · Reviewer_EncY · 2025-03-15
> **Comment**
>
> Thank you for the detailed response. My concerns are mostly addressed.
>
> However, i got one question regarding the result on llama3-70B. I find this result interesting but rather counter-intuitive. The perplexity number is 1e6 on WikiText but the model could still obtain quite high accuracy on commonsense reasoning task. Such high perplexity number should suggest that the model does contain any language modeling ability. Can the authors elaborate more on this contradiction?

---

> ### Author Response · Authors · 2025-03-20
> **Comment (1/1)**
>
> We appreciate the reviewer’s thoughtful question and the opportunity to clarify this point.
>
> We have used the LM Evaluation Harness [1] to evaluate tasks such as **BoolQ, MMLU, ARC-E, and ARC-C,** which rely on **log-likelihood** scoring to determine the correct answer. In these multiple-choice tasks, the model assigns log probabilities to each candidate answer, and the option with the **highest likelihood** is selected. This evaluation method primarily depends on the **relative ranking of the answer choices** rather than the absolute likelihood values.
>
> In contrast, perplexity is measured over full sequences and reflects the model’s overall ability to assign high probabilities to naturally occurring token sequences. Specifically, **perplexity is computed as the exponentiated average negative log-likelihood per token**, where lower perplexity indicates greater confidence in next-token prediction across the dataset.
>
> The observed contradiction—**high perplexity on WikiText but retained accuracy on reasoning tasks**—suggests that pruning has led to a **diffusion of probability mass across the vocabulary**. This means the model becomes **less confident in predicting the next token in an open-ended generation setting**, thereby increasing perplexity. However, in discriminative tasks like multiple-choice reasoning, where the model only needs to compare a limited set of completions, the **relative ordering of correct and incorrect answers is largely preserved**. Thus, despite higher perplexity, the model retains its ability to differentiate between correct and incorrect choices, maintaining strong performance on these reasoning benchmarks.
>
> [1] https://github.com/EleutherAI/lm-evaluation-harness

---

### Review · Reviewer_wMGw · 2025-02-05

**Summary Of Contributions:**

Authors evaluate width and depth pruned model on variety of tasks and show, that pruned models do not perform well on GSM8k and other harder benchmarks.

**Audience:**

No

**Broader Impact Concerns:**

None.

**Claims And Evidence:**

No

**Requested Changes:**

See weaknesses.

**Strengths And Weaknesses:**

Strengths:
Deeper evaluations are always valuable.

Weaknesses:
I do not like the writing; the authors spend half of the paper on the setup and introduction.

I also do not like the fact, that figure 1 and 4 are far away. Wikitext perplexity is the main measure used in evaluating pruned models. If you want to show, that this measure is not enough and we need to do deeper evaluations, showing lack of correlation between benchmarks and wikitext perplexity would be very valuable.

This statement from the abstract does not hold  "Unexpectedly, our analysis finds that depth pruning, despite its simplicity, consistently
outperforms the more granular width pruning approaches in maintaining downstream task performance."
Just look at Llama-3 numbers.

Also, I think the actual information provided is limited, and the analysis is simplistic. It would be great to see either the effects of fine-tuning or the performance of other methods (unstructured pruning/quantization). There are also long-range benchmarks; it would be interesting to see if performance depends on the task length.

---

> ### Author Response · Authors · 2025-02-27
> **Official Comment 1/2**
>
> We appreciate the reviewer’s feedback on our evaluation being valuable to the community.
>
> ### W1 :  I do not like the writing; the authors spend half of the paper on the setup and introduction.
>
> We would like to thank the reviewer for their suggestions. We have reduced the background required to explain the details of structured training-free pruning and moved some parts to the Appendix. This makes reading our work easier and aligns with the recommendation of the reviewer.
>
> ### W2: Wikitext perplexity is the main measure used in evaluating pruned models. If you want to show, that this measure is not enough and we need to do deeper evaluations, showing lack of correlation between benchmarks and wikitext perplexity would be very valuable.
>
> We agree with the reviewer that to strongly motivate our work, we can directly show a lack of correlation between perplexity and downstream task performance.
>
> From our experiments we find that depth pruning performs severely worse on perplexity compared to width pruning. However we find that when it comes to downstream reasoning tasks depth pruning usually outperforms width pruning. This in itself shows that perplexity alone cannot be a judge of model performance and other benchmarks. Additionally, even when perplexity is retained, downstream task performance - Coding, Mathematical Reasoning among others undergo severe performance degradation as shown in Figure 2, Figure 1 (GSM8k), Figure 4 in the main paper. These evidence substantiate our claim that perplexity measure is not enough to gauge the performance of pruned networks
>
> Finally, if we look at the difference between the performance of Depth Pruned and Width pruned models on the extended reasoning benchmarks as shown in Figure 3 we find that depth pruning having significantly larger perplexity at 30% compression across all models outperforms width pruning across all models in a clear majority of tasks with a exception for LLaMA3-8B MMLU where we find that width pruning is significantly better than depth pruning. We address this anomaly below.
>
>
>
> ### W3:  This statement from the abstract does not hold "Unexpectedly, our analysis finds that depth pruning, despite its simplicity, consistently outperforms the more granular width pruning approaches in maintaining downstream task performance." Just look at Llama-3 numbers.
>
> We thank the reviewer for pointing out this particular anomaly. We would like to redirect the reviewer to Figure 3 of the main paper. Except for MMLU, 30% pruning, LLaMA3-8B and GSM8k, 10% pruning, Qwen2-1.5B; depth pruning usually improves or is on par with width pruning across tasks and models. Further, we have changed our claim to - **Our analysis also finds that depth pruning, despite its simplicity, usually outperforms the more granular width pruning approaches in maintaining downstream task performance.** We hope that this aligns with the experimental observations and makes our work more aligned with the reviewer’s feedback.

---

> ### Author Response · Authors · 2025-02-27
> **Offiicial Comment 2/2**
>
> ### W4: It would be great to see either the effects of fine-tuning or the performance of other methods (unstructured pruning/quantization). There are also long-range benchmarks; it would be interesting to see if performance depends on the task length.
>
>
>
> Upon the recommendation of the reviewer, we have added experiments on unstructured pruning with both SparseGPT and Wanda on common sense reasoning benchmarks, mathematical reasoning (GSM8K), Code Eval and Instruction Following (IFEVAL). Our findings indicate that these unstructured methods, which offer finer granularity in pruning, better preserve performance on downstream tasks compared to structured - depth and width pruning approaches. The only benchmark where these methods show substantial deterioration is mathematical reasoning. This experiment provides further insights on how unstructured pruning affects LLMs and if the target hardware supports unstructured sparsity, these methods are a better choice as compared to structured - depth and width pruning approaches.
>
> **Results on SparseGPT and Wanda across varying budgets and downstream tasks.**
> | Pruning Method | Pruning (%) | ARC_E | ARC_C | Winogrande | BoolQ | MMLU  | GSM8k | CodeEval | WikiText | IFEVAL |
> |---------------|--------------|-------|-------|------------|-------|------|-------|---------|---------|--------|
> | Wanda        | 10           | 79.88 | 50.42 | 73.40      | 81.47 | 61.81 | 50.03 | 48.11   | 5.58    | 40.29  |
> | Wanda        | 20           | 79.71 | 49.14 | 72.61      | 81.87 | 61.11 | 47.83 | 49.90   | 5.71    | 39.18  |
> | Wanda        | 30           | 78.57 | 48.38 | 73.40      | 81.80 | 59.31 | 41.39 | 48.11   | 6.03    | 41.03  |
> | SparseGPT    | 10           | 80.38 | 50.34 | 72.92      | 81.71 | 61.69 | 47.91 | 49.14   | 5.58    | 41.22  |
> | SparseGPT    | 20           | 79.59 | 49.31 | 72.85      | 81.68 | 60.95 | 48.06 | 48.28   | 5.72    | 41.77  |
> | SparseGPT    | 30           | 78.70 | 47.61 | 74.11      | 81.89 | 59.88 | 43.59 | 47.93   | 6.05    | 39.74  |
>
>
> Additionally, we present evaluations on long-context benchmarks using the LLaMA3-8B-Instruct model within the LEval framework [1]. LEval includes multiple challenging tasks: QuALITY, which consists of questions based on long literary stories from Gutenberg; Coursera, designed to assess the reasoning ability of LLMs on complex and difficult-to-comprehend course materials; SFiction, which features questions that contradict real-world principles, testing the model’s ability to rely on input context rather than pretrained knowledge; and GSM100, a subset of 100 grade school math problems from the GSM8K dataset, aimed at evaluating few-shot learning capabilities in very long-context scenarios. The table below presents the performance of different pruning strategies at 10% and 20% compression on long-context benchmarks, compared to the Baseline. Depth pruning initially improves performance on SFiction but degrades on Coursera and GSM100 as pruning increases. Width pruning shows a trade-off, with 10% pruning slightly improving overall performance but 20% pruning significantly reducing accuracy, especially on GSM100. Overall in the case of long context understanding, we do not see a clear pattern with respect to performance degradation.
>
> **Results on LLaMA-3-8B-Instruct on long-context benchmarks across varying budgets**
>
> | Strategy  | Pruning % | Quality | Coursera | SFcition | GSM100 |
> |-----------|----------|---------|----------|---------|--------|
> | Baseline | -        | 62.38   | 52.33    | 69.53   | 80.00  |
> | Depth    | 10%      | 62.38   | 47.24    | 75.00   | 74.00  |
> | Depth    | 20%      | 62.87   | 42.88    | 67.19   | 39.00  |
> | Width    | 10%      | 63.86   | 44.19    | 70.31   | 65.00  |
> | Width    | 20%      | 60.40   | 51.45    | 67.97   | 39.00  |
>
> ### References
> [1] [https://arxiv.org/pdf/2307.11088](https://arxiv.org/pdf/2307.11088)

---

### Review · Reviewer_dot3 · 2025-02-12

**Summary Of Contributions:**

- Compared width vs. depth pruning of language models without finetuning
- Analyzed instruction-tuned models instead of just base models that have been the focus of prior work
- Showed that despite claims of retaining performance (such as ShortGPT claiming MMLU to be the retrained), performance is sacrificed when looking at a more holistic suite of tasks for evaluation

**Audience:**

Yes

**Claims And Evidence:**

Yes

**Requested Changes:**

- Page 3: "however the improvements with PEFT are inferior as compared to full-finetuning; especially in the case of pruned LLMs". Add citation to back this strong claim -- this paper is certainly relevant, but doesn't talk about pruning: https://arxiv.org/abs/2405.09673
- Weird heading at the end of page 8 ("Throughput Measurement of Depth and Width Pruned Models:") -- neither a paragraph header, nor a heading. Maybe make it a subsection? You don't seem to be short on space.
- Page 10: table should be set to be on the top rather than weirdly placed after a couple of lines.
- Include other methods for width pruning beyond the current FLAP method

**Strengths And Weaknesses:**

## Strengths

- The paper is well written
- The experiments are well designed and clear in terms of their conclusions
- The topic is important and relevant to a wider audience


## Weaknesses

- There have been some recent attempts to analyze the true impact of depth pruning (e.g., see https://arxiv.org/abs/2407.16286). They also showed that having information about the exact task being solved can be leveraged to further boost performance via a more careful selection of model blocks to prune. This makes the contribution in 4.3 not truly novel. The only aspect that I found novel is the fact that you attempted to look at chat-tuned models instead of just pretrained models, which makes it possible to compare on instruction-tuning benchmarks.
- Why not consider one shot approaches such as SparseGPT (https://arxiv.org/abs/2301.00774) for width pruning, which assumes the calibration set you already have access to for estimating importance? That seems to be aligned with the overall theme of the paper. While ShortGPT is close to the best methods we have for depth pruning, the literature on width pruning is significantly richer.
- There have been other attempts to construct useful metrics for importance estimation of different attention heads and neurons (e.g., see https://arxiv.org/pdf/2406.10260). Why didn't the authors compare against anything else beyond FLAP? Seems like an unfair comparison between depth and width. While there are only limited useful metrics for depth, a significant amount of work has gone into estimating the importance of parameters along the width dimension. I won't feel comfortable with such a strong general statement while looking at just one particular method.


## Questions

- Page 11; Fig 5a. Religion results with 10% depth pruning are worse than the baseline, but almost on par with the baseline at 30%. Seems weird. Any particular reason for that? This is also consistent with Fig. 7a.

---

> ### Author Response · Authors · 2025-02-27
> **Official Comment 1/2**
>
> We sincerely appreciate the reviewer’s thoughtful feedback and recognition of the strengths of our work, particularly in terms of the experiment design, clear conclusions and the broader impact of our findings.
>
> ### W1 : Why not consider one shot approaches such as SparseGPT ([https://arxiv.org/abs/2301.00774](https://arxiv.org/abs/2301.00774)) for width pruning, which assumes the calibration set you already have access to for estimating importance? That seems to be aligned with the overall theme of the paper. While ShortGPT is close to the best methods we have for depth pruning, the literature on width pruning is significantly richer.
>
>
>
> We thank the reviewer for pointing out areas where additional comparisons and clarifications could strengthen our contribution. We would like to point out the fact that the focus of this work is primarily on structured pruning, SparseGPT[1], Wanda[2] are two unstructured pruning methods and hence we do not include them in our existing evaluations. Further, upon the recommendation of the reviewer we have incorporated an evaluation over a set of unstructured pruning baselines. We have conducted experiments using both SparseGPT[1] and Wanda[2] on common sense reasoning benchmarks, mathematical reasoning (GSM8K), Code Eval and Instruction Following (IFEVAL). Our findings indicate that these unstructured/semi-structured methods, which offer finer granularity in pruning, better preserve performance on downstream tasks compared to structured - depth and width pruning approaches. The only benchmark where these methods show substantial deterioration is mathematical reasoning. This experiment provides further insights on how unstructured pruning affects LLMs and if the target hardware supports unstructured sparsity, these methods are a better choice as compared to structured - depth and width pruning approaches.
>
> **Results on SparseGPT and Wanda across varying budgets and downstream tasks.**
>
> | Pruning Method | Pruning (%) | ARC_E | ARC_C | Winogrande | BoolQ | MMLU | GSM8k | CodeEval | WikiText | IFEVAL |
> |---------------|---------|-------|-------|------------|-------|------|-------|---------|---------|--------|
> | Wanda | 10 | 79.88 | 50.42 | 73.40 | 81.47 | 61.81 | 50.03 | 48.11 | 5.58 | 40.29 |
> | Wanda | 20 | 79.71 | 49.14 | 72.61 | 81.87 | 61.11 | 47.83 | 49.90 | 5.71 | 39.18 |
> | Wanda | 30 | 78.57 | 48.38 | 73.40 | 81.80 | 59.31 | 41.39 | 48.11 | 6.03 | 41.03 |
> | SparseGPT | 10 | 80.38 | 50.34 | 72.92 | 81.71 | 61.69 | 47.91 | 49.14 | 5.58 | 41.22 |
> | SparseGPT | 20 | 79.59 | 49.31 | 72.85 | 81.68 | 60.95 | 48.06 | 48.28 | 5.72 | 41.77 |
> | SparseGPT | 30 | 78.70 | 47.61 | 74.11 | 81.89 | 59.88 | 43.59 | 47.93 | 6.05 | 39.74 |
>
> ### W2 : Seems like an unfair comparison between depth and width. While there are only limited useful metrics for depth, a significant amount of work has gone into estimating the importance of parameters along the width dimension. I won't feel comfortable with such a strong general statement while looking at just one particular method.
>
>
>
> We agree with the reviewer that depth pruning methods have a limited set of useful metrics for importance estimation, for example [3], [4] and [5] use similarity based importance estimation to prune blocks in the depth dimension. We have additionally added a comparative evaluation of LLM-Pruner[6] which is a competing width pruning method. However we find that our chosen width pruning method consistently outperforms LLM-Pruner. Moreover we have changed our claim to - **Our analysis also finds that depth pruning, despite its simplicity, usually outperforms the more granular width pruning approaches in maintaining downstream task performance.** which better reflects our experimental observations.
>
> | Method      | Pruning (%) | ARC_C | ARC_E | BoolQ | Winogrande |
> |------------|------------|-------|-------|-------|------------|
> | LLM-Pruner | 20%        | 37.18 | 59.18 | 59.39 | 61.33      |
> | FLAP       | 20%        | **39.25** | **69.91** | **69.63** | **68.85** |
> | LLM-Pruner | 50%        | 25.51 | 32.83 | 52.57 | 49.01      |
> | FLAP       | 50%        | **29.95** | **49.66** | **60.21** | **57.54** |

---

> ### Author Response · Authors · 2025-02-27
> **Official Comment 2/2**
>
> ### Q1 : Page 11; Fig 5a. Religion results with 10% depth pruning are worse than the baseline, but almost on par with the baseline at 30%. Seems weird. Any particular reason for that? This is also consistent with Fig. 7a.
> We sincerely thank the reviewer for this observation. In general, it has been noted that biases tend to decrease as language models undergo higher compression[7]. However, identifying the exact reason for the increase in bias at 10% depth pruning for this specific class is challenging, as pruning affects model representations in complex ways. We acknowledge this as an interesting phenomenon and appreciate the insight. We have added the corresponding reference in the main paper.
>
>
> ### R1 : Page 3: "however the improvements with PEFT are inferior as compared to full-finetuning; especially in the case of pruned LLMs". Add citation to back this strong claim -- this paper is certainly relevant, but doesn't talk about pruning: [https://arxiv.org/abs/2405.09673](https://arxiv.org/abs/2405.09673)
>
>
>
> We sincerely thank the reviewer for their suggestion and for providing a relevant reference. We have now included the mentioned citation in the paper to support our claim regarding the performance of PEFT compared to full fine-tuning. Additionally we have modified our claim to - “however the improvements with PEFT are inferior as compared to full-finetuning”. We could not find a related work to back our original claim regarding pruned LLMs, however we believe that PEFT methods require a strong base network to learn over, pruned LLMs are not strong base networks due to a drastic parametric elimination and hence without a full-parametric fine-tuning, PEFT cannot contribute much to the overall model performance. However this claim will require further experimentation which we feel is out of scope of this work.
>
>
>
> ### R2 : Weird heading at the end of page 8 ("Throughput Measurement of Depth and Width Pruned Models:") -- neither a paragraph header, nor a heading. Maybe make it a subsection? You don't seem to be short on space
>
>
>
> Thank you for your suggestion. We have revised the heading at the end of page 8 to be a proper subsection for better clarity and structure.
>
>
>
> ### R3 : Page 10: table should be set to be on the top rather than weirdly placed after a couple of lines
>
> We appreciate your feedback regarding the placement of the table on page 10. We have now adjusted its position to improve readability and alignment with the text.
>
>
>
> ### R4: Include other methods for width pruning beyond the current FLAP method
>
>
>
> We have now added a comparative evaluation of LLM-Pruner[6] which is a competing width pruning method. However we find that our chosen width pruning method consistently outperforms LLM-Pruner. Moreover we have changed our claim to - **Our analysis also finds that depth pruning, despite its simplicity, usually outperforms the more granular width pruning approaches in maintaining downstream task performance.**
>
> ### References
>
> [1] [https://arxiv.org/abs/2301.00774](https://arxiv.org/abs/2301.00774)
>
> [2] [https://arxiv.org/abs/2306.11695](https://arxiv.org/abs/2306.11695)
>
> [3] [https://arxiv.org/pdf/2403.17887](https://arxiv.org/pdf/2403.17887)
>
> [4] [https://arxiv.org/abs/2406.15786](https://arxiv.org/abs/2406.15786)
>
> [5] [https://arxiv.org/pdf/2402.11187](https://arxiv.org/pdf/2402.11187)
>
> [6] [https://arxiv.org/abs/2305.11627](https://arxiv.org/abs/2305.11627)
>
> [7] [https://aclanthology.org/2023.acl-long.878/](https://aclanthology.org/2023.acl-long.878/)

---

> > ### Comment · Reviewer_dot3 · 2025-03-18
> > **Thank you for the detailed responses**
> >
> > I am thankful to the authors for taking out the time in doing the experiments, and for the detailed responses.
> >
> > [W1] I didn't find any info on the model used to perform these experiments, and they were just buried in the appendix. While it is certainly upto the authors, the only reason why I thought this was interesting was to make the paper more useful to the readers. I would suggest to do a good job and polish the results in terms of their presentation and integrate them in the main paper.
> >
> > [W2] Why aren't these results part of the paper? Again, my aim is to just provide external feedback on how to improve the paper. I think these results would be useful to a reader, not just to satisfy my random request.
> >
> > ---
> > Lastly, I didn't see any response from the authors regarding weakness 1 in my review, which was my only true concern as it claims credit for something which is not truly novel anymore. While the other paper isn't that old (which I agree is a concern), but should have been incorporated to indicate that this wasn't the first attempt to do this (need to give credit where it is due).
> >
> > Thanks again for all the effort!

---

> > > ### Author Response · Authors · 2025-03-23
> > > **Comment (1/1)**
> > >
> > > We appreciate the reviewer’s insightful feedback on the presentation of our results and their relevance to a broader audience.
> > >
> > > **[W1] I didn't find any info on the model used to perform these experiments, and they were just buried in the appendix. While it is certainly upto the authors, the only reason why I thought this was interesting was to make the paper more useful to the readers. I would suggest to do a good job and polish the results in terms of their presentation and integrate them in the main paper.**
> > >
> > >
> > > **[W2] Why aren't these results part of the paper? Again, my aim is to just provide external feedback on how to improve the paper. I think these results would be useful to a reader, not just to satisfy my random request.**
> > >
> > > Taking this into account, we have integrated the results mentioned in W[1] and W[2] of the original review into the main paper and ensured that experimental details are included.
> > >
> > > **Lastly, I didn't see any response from the authors regarding weakness 1 in my review, which was my only true concern as it claims credit for something which is not truly novel anymore. While the other paper isn't that old (which I agree is a concern), but should have been incorporated to indicate that this wasn't the first attempt to do this (need to give credit where it is due).**
> > >
> > >
> > > Apologies for missing weakness 1 of the review. We have cited the mentioned paper in section 4.3 as our findings support the performance drop in depth-pruned models. Moreover, in our original paper we have extended it towards width-pruning and evaluated on other benchmarks (IFEVAL, HumanEval) as well.

---

### Decision · Action_Editor_UanH · 2025-03-25

**Recommendation:** Accept as is

**Comment:**

The paper provides a detailed analysis of training-free structural pruning methods for LLMs and highlights shortcomings in their evaluation within the literature.

Reviewers acknowledge that the analysis is generally accurate, though minor improvements could be made. Reviewer wMGw noted that the paper focuses only on small pruning rates, which may not yield significant practical gains, and suggested that additional pruning methods could be further explored. However, in my view, the paper does not aim to provide a large-scale study of existing pruning methods but rather to highlight current limitations in the field.

Overall, I believe the paper has the potential to improve current practices and guide future research.

**Audience:**

Structural pruning is an efficient method for LLM compression and is certainly relevant to a portion of the TMLR audience.

**Claims And Evidence:**

The paper argues that existing evaluations of training-free structured pruning methods for LLMs often overstate their effectiveness and that depth pruning frequently outperforms width pruning on downstream tasks. To support these claims, it provides a detailed analysis across additional datasets and tasks.